# Hollow core optical fibres with comparable attenuation to silica fibres between 600 and 1100 nm

Hesham Sakr [1,3], Yong Chen[1,2,3], Gregory T. Jasion [1], Thomas D. Bradley [1], John R. Hayes[1], Hans Christian H. Mulvad[1], Ian A. Davidson[1], Eric Numkam Fokoua[1] & Francesco Poletti [1✉]

For over 50 years, pure or doped silica glass optical fibres have been an unrivalled platform for the transmission of laser light and optical data at wavelengths from the visible to the near infra-red. Rayleigh scattering, arising from frozen-in density fluctuations in the glass, fundamentally limits the minimum attenuation of these fibres and hence restricts their application, especially at shorter wavelengths. Guiding light in hollow (air) core fibres offers a potential way to overcome this insurmountable attenuation limit set by the glass's scattering, but requires reduction of all the other loss-inducing mechanisms. Here we report hollow core fibres, of nested antiresonant design, with losses comparable or lower than achievable in solid glass fibres around technologically relevant wavelengths of 660, 850, and 1060 nm. Their lower than Rayleigh scattering loss in an air-guiding structure offers the potential for advances in quantum communications, data transmission, and laser power delivery.

---

[1] Optoelectronics Research Centre, University of Southampton, Highfield Campus, SO17 1BJ Southampton, UK. [2] Lumenisity Ltd, Unit 7, The Quadrangle, Southampton SO51 9DL, UK. [3] These authors contributed equally: Hesham Sakr, Yong Chen. ✉email: fp@soton.ac.uk

Guiding light inside glassy optical fibres is an enabling technological platform that supports a $40B global cable market, with over 400 million km of fibre produced every year. Key applications are in high-speed data transfer—enabling the global internet and cloud-based data services, sensing for oil and gas installations, structural monitoring for railways and bridges, as well as medical, biophotonics, and defence areas.

Since Charles Kao's pioneering work in 1966[1] and the first demonstration of a sub 20 dB km$^{-1}$ optical fibre by Corning in 1970[2], scientists worldwide have searched for ways to reduce the propagation loss in glass-based light waveguiding technology, as a way to further expand its reach and impact.

All transparent glasses present a broad spectral window in the visible and near infra-red, between their ultraviolet (electronic) and infra-red (vibrational) absorption peaks. Since pioneering works in the 1970–80 s managed to adequately suppress absorptions from metallic impurities and hydroxyl overtones by reducing impurities in the glass to part per billion levels[3], the remaining loss in this spectral region is intrinsically caused by scattering due to local density fluctuations, Rayleigh scattering (RS)[3,4]. For decades thereafter, researchers have investigated drawable glass compositions offering the lowest possible Rayleigh scattering coefficient (RSC).

Since the RSC of a glass is proportional to the temperature at which the density fluctuations in the liquid state are frozen-in, low-softening temperature glasses, such as chalcogenides or multi-component heavy metal halides, have been studied thoroughly[5]. Although these can in principle offer RSC close to an order of magnitude lower than that of silica glass[6], the lowest loss produced to date in these glass families is still about 1000 times higher than their theoretically anticipated Rayleigh scattering limit. This is mostly due to seemingly insurmountable difficulties in the purification of the raw materials and to the tendency of these glasses to re-crystallise during the fibre draw process.

This has left single component silicon dioxide—silica—a glass that can be synthesised with the required part per billion purity through vapour phase deposition techniques and that does not suffer from re-crystallisation to the same degree, as the unbeatable go-to glass for state-of-the-art loss at wavelengths between ~300 and ~1700 nm. No other optical waveguide technology has been able to offer a lower propagation loss than pure or doped silica fibres. Decades of technological progress have brought the loss of these fibres very close to fundamental intrinsic limits, to the point that despite large research investments worldwide, this has only decreased by 0.015 dB km$^{-1}$ in the last 35 years[7]. Applications have thus been forced to progress around the seemingly fundamental attenuation limit set by Rayleigh scattering. Due to the $\lambda^{-4}$ dependence of RS, while the minimum absolute loss achieved at 1550 nm is only 0.142 dB/km[7], the lowest reported loss at the shorter technologically relevant wavelengths of 1060, 830, and 630 nm increase to 0.57, 1.6, and 4.5 dB km$^{-1}$ respectively. These record-low loss results can only be achieved in research-grade phosphosilicate fibres[8], or in fibres with a pure silica core and a down-doped Fluorine-containing cladding[7,9]. The latter is drawn slowly (hence in small volumes) to reduce their fictive temperature[10], and mostly used in submarine applications, a small subset of the total market. The typical Rayleigh scattering loss of the Germanium-doped core fibres that form the bulk of the fibre optic market today is about 30% higher[11].

This small but finite loss imposes limitations on the achievable transmission lengths over which optical fibres can be practically employed, again, especially at shorter wavelengths. Furthermore, in addition to loss, the use of silica core fibres in the visible/near infra-red also suffers from other intrinsic problems. For example, having a sufficiently small core so that a single transverse mode propagates in the fibre is preferable in most circumstances, since it avoids

intermodal dispersion and increases temporal and spatial stability. However, ensuring single-mode operation at shorter wavelengths requires a proportionally smaller core, making more expensive coupling solutions with tighter alignment tolerances necessary, which often increase the losses at launch. The fibre's chromatic dispersion also increases – which is detrimental for pulse and data transmission – as one moves away from the zero-dispersion wavelength of silica at ~1300 nm towards its UV absorption peak at 195 nm[12]. Finally, all-glass optical fibres suffer from optical nonlinearities and ultimately dielectric breakdown above a certain peak incident power, and present an intrinsic sensitivity to external temperature and mechanical perturbations[13,14]. All these factors combine to limit the achievable performance in solid-core fibres, and hence their application.

Since the seminal work by Lord Rayleigh, who in 1897 derived the theoretical formalisms for the transmission of electromagnetic waves in hollow dielectric structures[15], waveguiding light in air or vacuum has been studied as a potential alternative to transmitting light through transparent solid-state media. The lower density of gases decrease their RSC, and the related loss contribution, by 2–3 orders of magnitude, but light guidance in air is not trivial and other loss mechanisms came into play. It was only through the invention of photonic bandgap materials in the 1990s, and their application to waveguiding in an optical fibre[16,17], that flexible dielectric hollow-core fibres (HCFs) became a potential contender in the race for the lowest possible optical loss.

In the past 20 years, hundreds of studies have been devoted to understanding the physical guidance mechanism in HCFs and optimising their structure to minimise loss and improve their modal properties, with the aim of beating the loss limit of solid glass waveguides. Initial HCFs exploited out-of-plane photonic bandgaps to waveguide light in a low refractive index core region[17,18]. More recently, the attention of researchers worldwide has shifted to HCFs that exploit a combination of inhibited coupling between air and glass modes[19] and antiresonance from core surrounding membranes[20] to achieve improved optical performance.

These fibres now exploit core surrounding membranes with a negative curvature[21] and with a nodeless glass design[22] to minimise the overlap of light with the glass cladding and reduce leakage loss. With this approach, nodeless 'tubular' or 'revolver' antiresonant HCFs have been reported with losses as low as 13.8 dB km$^{-1}$ at 539 nm[23] and 7.7 dB km$^{-1}$ at 750 nm[24]. Crucially, loss in these structures is no longer dominated by Rayleigh scattering, but rather by the leakage of light out of the core. Numerous designs have been therefore proposed to reduce light leakage. One of the most successful concepts so far has proven the addition of smaller nested tubes to increase the number of light confining air-glass interfaces, in hollow-core Nested Anti-resonant Nodeless Fibres (NANFs)[25,26]. In recent experimental demonstrations, these fibres have significantly closed the loss gap to conventional fibres at telecoms wavelengths, with a reported loss of 1.3 dB km$^{-1}$ in 2018[27], 0.65 dBkm$^{-1}$ in 2019[28], and 0.28 dB km$^{-1}$ in 2020[29].

An advantage of all these antiresonant hollow-core fibres, in which Rayleigh scattering plays a negligible contribution, is that their remaining loss mechanisms have a more favourable wavelength scaling than $\lambda^{-4}$. They should therefore be more suitable for low loss operation at shorter wavelengths. However, optimising their guidance towards the visible range of the spectrum is more challenging since this requires scaling down the membrane thickness, which makes the rheology in the fibre draw process more difficult to control[30].

Despite this, very promising results have been recently achieved by exploiting a derivation of the NANF concept that uses straight membranes rather than nested tubes—the

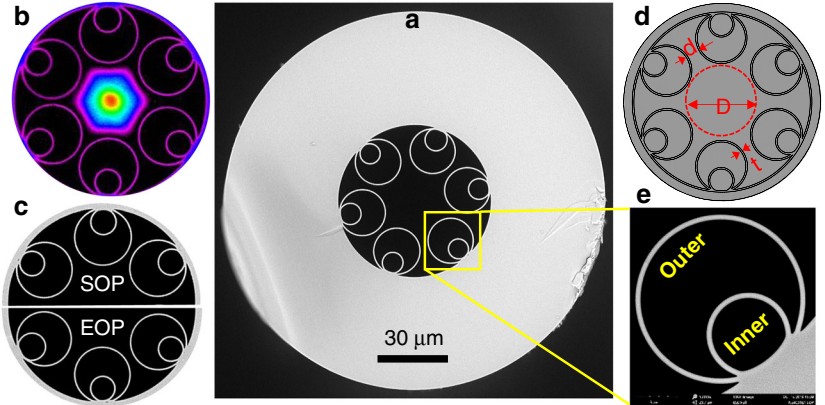

**Fig. 1 Nested Anti-resonant Nodeless Fibre (NANF). a** scanning electron micrograph cross-section of a typical fabricated hollow-core NANF.
**b** microscope image of the transmitted optical mode, confined in the air core surrounded by the 6 nested tubes. **c** comparison between the start of pull (SOP) and end of pull (EOP) cross-sections of the fibre, showing the uniformity achieved along the as-drawn fibre length**. d** and **e** illustrate the terms used in defining the structure.

**Table 1 Key optical and physical parameters of the fabricated NANFs including fibre length, loss at centre wavelength ($\lambda_c$), diameter of the microstructure, core diameter (D), inter-tube gaps (d), silica tube thickness (t) and outer diameter (OD).**

| NANF | Fibre length | $\lambda_c$/window | Optical properties | | Structural properties | | | | | | |
|---|---|---|---|---|---|---|---|---|---|---|---|
| | | | loss at $\lambda_c$ | 3 dB bandwidth | Structure diameter | Core OD (D) | Gaps (d) | Outers | | Inners | |
| | | | | | | | | OD | t | OD | t |
| Unit | m | nm | dBkm$^{-1}$ | nm | µm | µm | µm | µm | nm | µm | nm |
| A: SOP | 406 | 1064/2nd | 0.52 ± 0.05 | 176 | 83.8 | 32.5 | 2.4–5.1 | 25.6 | 740 | 12.1 | 800 |
| A: EOP | | | | | 84.2 | 33.3 | 2.5–5.5 | 25.6 | 750 | 12 | 810 |
| B: SOP | 1060 | 850/2nd | 1.45 ± 0.15 | 196 | 64.6 | 28.5 | 4.7–5.4 | 18.5 | 600 | 9.3 | 590 |
| B: EOP | | | | | 65.4 | 28.1 | 4.4–4.9 | 19 | 580 | 9.5 | 570 |
| C | 823 | 660/3rd | 2.85 ± 0.2 | 132 | 80 | 30.4 | 1.6–3.6 | 25.1 | 780 | 11.9 | 830 |

conjoined-tube negative curvature HCFs[31]. Using this concept, HCFs have been reported with minimum losses of 2 dB km$^{-1}$ at 1512 nm[31], 3.8 dB km$^{-1}$ at 680 nm[32], and 4.9 dB km$^{-1}$ at 558 nm[32], the very first report of a lower than silica Rayleigh scattering loss.

In this work, we further advanced the fabrication process of NANFs, and through a careful design to shift their operation to shorter wavelengths we have produced air-guiding fibres that beat the RS limit of silica glass at selected wavelengths between 600 and 1100 nm. Since over 99.99% of light is guided in air, these hollow-core NANFs also solve many of the other problems affecting solid-core fibres. The fibres are effectively single moded, despite large core diameters 6 to 8 times those of comparable single-mode solid core fibres[27]. Besides, they exhibit a near-zero chromatic dispersion at all low-loss wavelengths, considerably lower sensitivity than glass fibres to changes in the external environment[33–35], and predicted >1000 times lower optical nonlinearity[36] and higher laser damage threshold[37].

## Results

**Fibre description and characterisation.** Figure 1 shows an example of a fabricated NANF. Here, a central microstructured region contains an air core, typically as wide as 25–50 optical wavelengths in vacuum, surrounded by 6 nested cylindrical glass membranes that are attached to the inside of a glass jacket tube, Fig. 1a. The thickness of the 6 inner and outer tubes is nominally

matched for optimum operation, and is carefully controlled to create an antiresonant condition in the transverse plane that confines light effectively in the air core, and simultaneously reduces the light overlap with the glass[25]. Through an embedded high-order mode stripping mechanism, the fibre supports a single spatial mode, as shown in Fig. 1b and characterised experimentally in other works[27–29,38]. Despite the seemingly fragile structure, the fibres are fabricated with stable and highly controllable processes which make cross sections almost indistinguishable between start of pull (SOP) and end of pull (EOP), as in the example shown in Fig. 1c. The definition of core diameter (D), inter-tube gaps (d), wall thickness (t), primary cladding tube ('outer'), and nested cladding tube ('inner') is shown in Fig. 1d, e.

We have fabricated three NANFs using the stack, fuse, and draw method, with selective pressure differentials during the draw allowing control of the final geometry (see methods). The fibres are named *NANF-A, B*, and *C*, and have lengths of 406 m, 1060 m, and 823 m, respectively. Their main structural properties are summarised in Table 1, and their cross-sections are shown in Fig. 2.

In NANFs, like in other anti-resonant fibre (ARF) types, the thickness of the cladding tube membrane (high-index layer) determines the spectral position of the high loss resonances and the operational wavelength range in between where low loss guidance can be achieved, according to the antiresonant reflecting optical waveguide (ARROW) model[20,39]. Each high-index glass layer acts as a Fabry-Perot resonator in the plane transverse to the

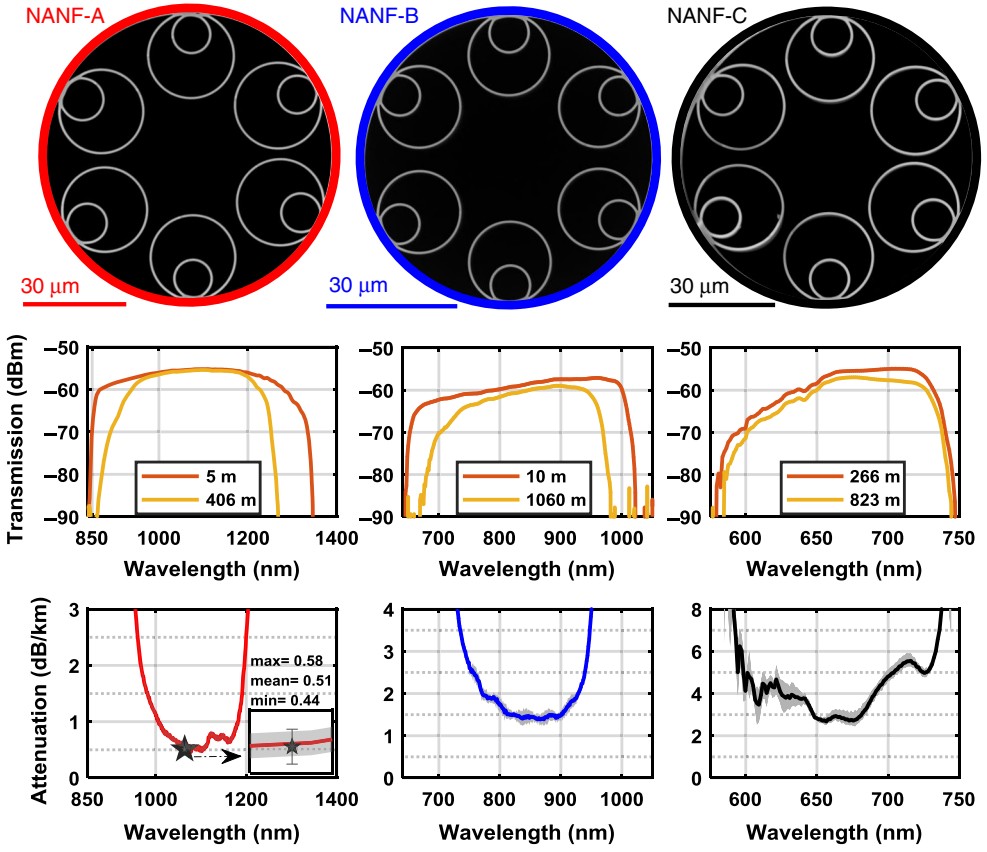

**Fig. 2 Structure and transmission properties of the reported fibres. Top**, SEM images illustrating the structure. **Middle**, transmission spectra of long and short lengths of the fibres averaged over three different fibre cleaves. **Bottom**, measured cutback spectra at the relevant antiresonance passband. The shaded area in grey represents the uncertainty in the cutback measurements (see Methods).

fibre axis and contributes to maintain the air-guided modes confined within the core. The specific design of the structured cladding (tube diameter and inter-tube azimuthal spacing) determines the overall low loss bandwidth[24], while the radial distance between inner and outer tubes controls the amount of additional leakage loss observed by the higher-order modes[20,25].

Designing a fibre that operates in the 1st antiresonant window (i.e., at wavelengths longer than $\lambda_1 = 2t\sqrt{n^2 - 1}$) confers it a wider bandwidth and higher yield compared to a fibre that guides in higher-order (2nd or greater) windows, but requires thinner tubular membranes, making controlling the rheology of the fabrication process more difficult[40]. While 1st window NANFs with excellent structural symmetry have recently been demonstrated for 1550 nm operation, shifting their operation to shorter wavelengths is more challenging from a fabrication perspective as it requires smaller fibres with a larger draw down ratio which accentuate asymmetries in the original preform. Besides, the smaller dimensions in the targeted tubes lead to thinner membranes that offer a lower viscous resistance to counteract the hole-collapsing tendency of surface tension, which is also higher[30].

Therefore, we decided to design and fabricate *NANF-A* and *B* so that they operated in the 2nd window at 1064 and 850 nm, respectively. *NANF-A* has outer tubes with an average membrane thickness of 745 ± 5 nm (first resonance at $\lambda_1 = 1.56$ μm), whereas *NANF-B* has thinner membrane thicknesses of 590 ± 10 nm (first resonance at $\lambda_1 = 1.24$ μm). While in all cases we aimed for equal membrane thickness between inner and outer tubes (Fig. 1e), for *NANF-A* and *B* we achieved inner tubes on average 10% thicker and 5% thinner, respectively. This variation is representative of

current fabrication processes and leaves margins for improvement. The core diameter of both fibres was chosen to provide a ~30% larger diameter to operational wavelength ratio ($D/\lambda$) than in fibres operating at 1550 nm, and is ~33 and ~28 μm, for *NANF-A* and *B* respectively. As can be appreciated from their cross-section, all fabricated fibres present a high degree of symmetry, which is key to their reduced leakage loss. For example, *NANF-B* (SOP) has tubes azimuthally positioned at angles ranging from 59.1° to 61.8°, outer tube OD varying from 18.4 to 18.7 μm, and inner tube OD varying from 9.2 to 9.5 μm.

As a result of the practical challenges in producing and controlling thinner membranes, the achieved average gaps between its outer tubes ('d'; Fig. 1d) in *NANF-B* are larger than those in *NANF-A* (4.8 μm compared to 2.8 μm), which is sub-optimum for leakage loss control. In order to produce a fibre guiding at even shorter wavelengths, therefore, rather than thinning the tube membranes further, we opted to use the 3rd antiresonant window. *NANF-C* has outer tube membrane thicknesses of 780 ± 20 nm (similar to *NANF-A*) and a 3rd window centred at 660 nm wavelength. To make the fibre more resilient to microbend loss which is stronger at shorter wavelengths, we reduced the core diameter D by 3 μm ($D/\lambda$ ratio of ~45) and significantly decreased the microstructure to outer glass diameter ratio from 40% to 23.5% for improved mechanical stiffness[41,42]. Additionally, the fibres were coated with commercially available low modulus UV- curable polymers. The membrane thickness, tube and gap size of *NANF-C* are very comparable to those of *NANF-A* (see Table 1).

For all three fibres the structure is maintained throughout the drawn length, with less than two percent variations in key

geometrical parameters from SOP to EOP (see the images in Fig. 1c referring to *NANF-A*, and Table 1).

The measured optical transmission spectra and the attenuation of each fibre obtained via cutback (see methods) are shown in Fig. 2. *NANF-A* transmits across the wavelength range 950–1207 nm, with a loss of 0.52 ± 0.05 dB km$^{-1}$ at 1064 nm and 0.5 ± 0.05 dB km$^{-1}$ at 1100 nm, respectively. The fibre has a 3 dB bandwidth of 176-nm (defined as the bandwidth at which the loss in dB km$^{-1}$ doubles from its minimum value), with loss ≤ 1 dB km$^{-1}$ between 1006–1182 nm. It has a 230-nm region with loss ≤ 2 dB km$^{-1}$, between the wavelengths 966–1196 nm. The small peak at ~1120 nm is due to water vapour absorption in the hollow core resulting from the preform processing[43], and can be eliminated via dry gas purging during or after fibre draw. An additional cutback measurement on the same fibre using a laser diode operating at 1064 nm gave a loss value of 0.51 ± 0.07 dB km$^{-1}$ at 1064 nm (marked 'star' in the loss spectrum of *NANF-A* in Fig. 2), in good agreement with the broadband measurement. *NANF-B* has a loss ≤ 5 dB km$^{-1}$ from 722 to 955 nm reaching a minimum of 1.4 ± 0.15 dB km$^{-1}$ at 862 nm. The fibre has a substantially flat spectral transmission between 810–900 nm, with only ± 0.05 dB km$^{-1}$ loss variation across this region, and at 850 nm it has a loss of 1.45 ± 0.15 dB km$^{-1}$. Its 3 dB bandwidth (loss ≤ 2.8 dBkm$^{-1}$) spans the wavelength range 746–942 nm. Finally, *NANF-C* operates in the visible spectrum and has optical losses ≤ 7 dB km$^{-1}$ across the wavelengths range 593–735 nm. Its lowest loss of 2.85 ± 0.2 dB km$^{-1}$ is achieved at 660 nm, and its 3 dB bandwidth is 132-nm, extending from 598 to 730 nm.

To contextualise the loss of these three NANFs, in Fig. 3 we plot them in the same graph with state-of-the-art reported loss values from other optical fibre technologies. The shaded yellow band represents the fundamental Rayleigh scattering loss of state-of-the-art solid-core fibres. Its lower limit (black dashed line) represents what is achievable with pure silica core technology (PS1 is from Nagayama et al.[9], while PS2 from Tamura et al.[7] is the current record low loss fibre at 0.1424 dBkm$^{-1}$). Their

Rayleigh scattering coefficient of ~0.75 dB km$^{-1}$ μm$^{-4}$ is a result of 'cold draws' aimed at reducing the fictive temperature[10] and use of low scattering core glasses such as pure silica with or without traces of Fluorine[44]. As a drawback, the fibres typically need to be drawn at slower speeds to allow structural relaxation to occur[10], and are thus not representative of typical high-volume production fibres. The upper bound of the shaded area (black dashed curve) has a Rayleigh scattering coefficient of 1 dB km$^{-1}$ μm$^{-4}$, which is compatible with the loss experienced by high volume, fast draw speed, state-of-the-art commercial germanium doped silica fibres (loss at 1310 nm and 1550 nm below 0.35 and 0.20 dB km$^{-1}$, respectively[11]). On a similar low scattering curve as the pure silica core fibres are two multimode phosphosilicate fibres produced in the 1970s (Ph1, Horiguchi et al.[45]). Despite theoretical papers predicting that Rayleigh scattering coefficients as low as 0.64 dB km$^{-1}$ μm$^{-4}$ would be possible by further optimising the concentration of phosphorus doping in the glass[46], no experimental validation has yet been achieved.

Loss values in the shaded area are routinely achieved by commercial fibres at telecommunication wavelengths between 1300 and 1550 nm. This is however not the case at shorter wavelengths, as shown by the four yellow circular markers indicating the typical loss of commercial solid core fibres between 600 and 1060 nm (C1: single-mode fibre at red visible wavelengths[47]; C2: endlessly single-mode photonic crystal fibre LMA-5[48]; C3: highly multimode graded-index OM5 fibre for datacoms[49]; C4: single-mode high numerical aperture fibre for 980 nm pump delivery[50]). As can be seen, their loss can be substantially higher than the theoretical value. This comes potentially from the close vicinity of the UV absorption tail[3], or from a high Germanium concentration in the core (causing higher RS), needed to either tailor the fibre's intermodal dispersion at 850 nm, or to enhance numerical aperture at 1060 nm. The figure also shows the current state-of-the-art loss in hollow-core fibres (blue diamond markers). Fibres H1-H4 (references Chafer et al.[51], Debord et al.[52], Debord et al.[24], Chafer

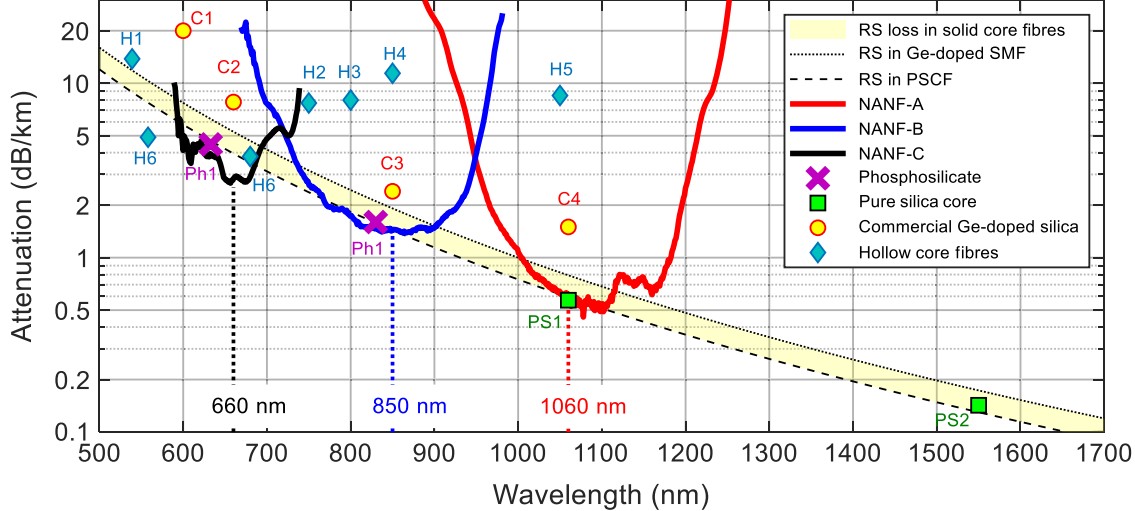

**Fig. 3 Loss of NANFs reported in this work.** *NANF-C* operating at 660 nm (solid black), *NANF-B* operating at 850 nm (solid blue), and *NANF-A* operating at 1060 nm (solid red). The shaded yellow band represents the fundamental Rayleigh scattering loss of state-of-the-art solid-core fibres, its lower limit (black dashed line) represents what is achievable with pure silica core technology (PS1 is from Nagayama et al.[9], while PS2 is from Tamura et al.[7]). The upper bound of the shaded area (black dotted curve) is compatible with state-of-the-art commercial germanium doped silica fibres[11]. On a similar low-scattering curve are two multimode phosphosilicate fibres produced in the 1970s (Ph1, Horiguchi et al.[45]). The four yellow circular markers indicate the typical loss of commercial solid core fibres between 600 and 1060 nm (C1: single-mode fibre at red visible wavelengths[47]; C2: large-area single-mode photonic crystal fibre LMA-5[48]; C3: highly multimode graded-index OM5 fibre for datacoms[49]; C4 single mode high numerical aperture fibre for 980 nm pump delivery[50]). The blue diamond markers represent the current state-of-the-art loss in hollow-core fibres - H1: Chafer et al.[51]; H2: Debord et al.[52]; H3: Debord et al.[24]; H4: Chafer et al.[23]; H5: Maurel et al.[53]; and H6: Gao et al.[32].

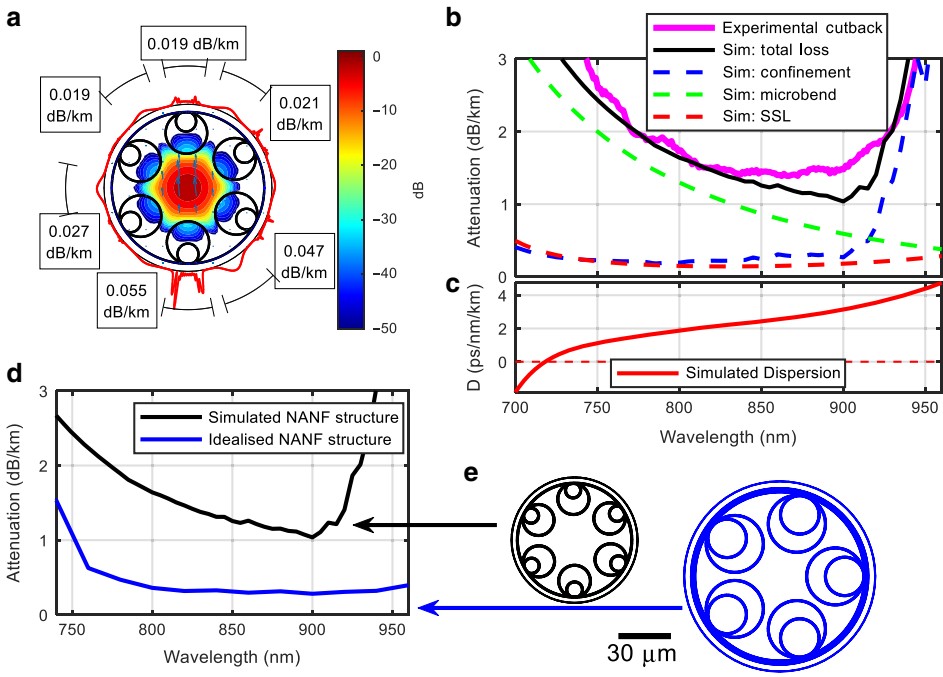

**Fig. 4 Simulation results of *NANF-B*. a** Fundamental mode at 850 nm with contours of longitudinal Poynting vector, and azimuthally resolved power leakage (radial Poynting vector) is visualised as a red external curve - local regions of high leakage are highlighted and quantified in the labels. **b** Measured fibre loss compared with total simulated loss (black) as the sum of the confinement loss (blue), surface scattering (red), and microbend (green) losses. **c** Group velocity dispersion of the simulated structure. **d** The total loss of the simulated *NANF-B* (black) compared with an idealised 5-tube structure (blue), depicted to scale in **e**.

et al.[23], respectively) have all a 'tubular' single ring structure and loss in the 7–15 dB km$^{-1}$. Fibre H5 (Maurel et al.[53]) has a Kagome structure and a loss of 8.5 dB km$^{-1}$, while the two fibres in H6 (Gao et al.[32]) have a lower loss thanks to the addition of a nested membrane associated with the conjoined tubular structure.

The cutback loss of our three fibres is overlapped on the same figure. In the wavelength range 1050–1100 nm, of interest to high power laser delivery applications, *NANF-A* shows a similar attenuation to the ultimate silica Rayleigh scattering loss (RSL), with losses down to 0.5 ± 0.05 dB km$^{-1}$. Compared to the few-moded pure silica core fibre in PS1 though, our fibre is effectively single-mode despite a mode field diameter nearly twice as large (~22.4 μm), and is further predicted to have three to four orders of magnitude higher damage threshold. Besides, its loss is only 35% that of commercial silica fibres at these wavelengths (C4, Corning HI 1060 with attenuation of 1.5 dB km$^{-1}$ at 1060 nm[50]).

The spectral region around 850 nm is also of high technological interest as it encompasses the operational window of the Ti-sapphire laser (the workhorse of ultrafast optics), of low cost high-speed GaAs VCSELs and of highly efficient, low-noise silicon photodetectors. At these wavelengths, the loss of *NANF-B* (~1.45 dB km$^{-1}$ between 810 and 900 nm) is comparable to the RSL of pure silica, and at 830 nm it is marginally lower (by 0.1 dB km$^{-1}$) than the lowest loss solid fibre to our knowledge, Ph1. Furthermore, the loss of *NANF-B* remains below the germano-silicate RSL limit from 730 to 900 nm, and it is only 60% that of commercial multimode graded-index fibres widely used in enterprise and datacentre datacoms applications (e.g., Corning OM$_{2-5}$ with attenuation ≤2.3 dB km$^{-1}$ at 850 nm[49]). This effectively single moded NANF also shares the same ease of interconnection as OM few moded graded-index fibres thanks to its large core size and MFD, but it presents a chromatic dispersion of 2–3 ps nm$^{-1}$ km$^{-1}$ (shown later in Fig. 4c), significantly below the ~100 ps nm$^{-1}$ km$^{-1}$ of the solid core counterpart.

In the visible range of the spectrum, used extensively for biophotonics and quantum applications, *NANF-C*'s measured loss of 2.85 ± 0.2 dB km$^{-1}$ at 660 nm is 71% lower than the ultimate RSL of silica, ~4 dB km$^{-1}$. In this spectral region, the loss of solid-core fibres is also beginning to be affected by ultraviolet absorptions in the glass, which can in part explain the 20 dB km$^{-1}$ of commercial single-mode fibre at 600 nm (C1, Corning RGB 400[47]). The estimated mode field diameter of *NANF-C*, ~21 μm, will also greatly simplify the alignment tolerances required for coupling in and out of the fibre, as compared to a single-mode solid core fibre with a mode diameter of 3.9 ± 0.5 μm at 600 nm[47].

**Loss contributions and improved designs**. Losses in HCFs originate from intrinsic sources (e.g., leakage, surface scattering) and extrinsic ones (e.g., micro and macrobending), as well as from technological imperfections such as small structural variations along the length[54]. To understand the relative contribution of these different mechanisms, we show in Fig. 4 a detailed modelling analysis of *NANF-B* (see Methods for more information). Similar considerations apply to the other two fibres. Simulations indicate that in NANF-B the dominant loss mechanism in the wavelength range 700–910 nm arises from microbending effects due to the relatively large core to wavelength ratio. This decreases from 3 to ~0.5 dB km$^{-1}$ as the wavelength increases, as shown in Fig. 4b. Confinement loss is generally small (0.15–0.3 dB km$^{-1}$), but it increases substantially near the resonances, while surface scattering loss (SSL) only accounts for <15% of the total. Macrobend losses for a fibre spooled on a 30 cm diameter bobbin are predicted to be around 0.1 dB km$^{-1}$. The total simulated loss obtained by adding all these contributions is in substantial agreement with the measured loss, apart from small deviations in the 850–920 nm range. Figure 4a plots the azimuthally-resolved rate of power leakage through a cross-section of the fabricated fibre at 850 nm, and shows that due to a good symmetry and small inter-tube gaps the rate of leakage is low for every azimuthal

position, with no geometric features contributing more than 25% to leakage loss.

In Fig. 4c we report the calculated dispersion of the fibre, which crosses zero at 718 nm and remains below 3 ps nm$^{-1}$ km$^{-1}$ in the region of the minimum loss. For comparison, the material dispersion of silica at 850 nm is over 30 times larger, ~100 ps nm$^{-1}$ km$^{-1}$[55].

Despite the record-low loss measured in these 3 NANFs, it is worth stressing that these loss values are still far from fundamental limits, and considerable further reductions can be expected with refined designs. Increasing the fibre stiffness and decreasing the Young's modulus of its coating, for example, can reduce the contribution from microbending, while enlarging its core can reduce both SSL and confinement loss. Additional reductions can be achieved by considering alternative designs. As an example of what could be feasible, in Fig. 4d, e we show the loss of a design with 5 rather than 6 nested tubes, simulated with the same tools. As a result of larger individual tubes, smaller gaps (4 μm vs ~5 μm), a slightly enlarged core diameter (30 μm vs 28.5 μm), and a thicker outer cladding to reduce microbend contributions, its total loss at 850 nm is predicted to be as low as 0.3 dB km$^{-1}$, nearly ten times lower than the RS loss of silica.

## Discussion

In this work, we have shown that through greater understanding of the loss mechanisms in hollow-core fibres, choice of an appropriate structure—the NANF—careful control of design parameters and fabrication procedures, air-guiding fibres with loss comparable to, and in some cases lower than achievable in a solid core fibre can be realised at a range of wavelengths between 600 and 1100 nm.

When the loss values of the three NANFs presented here, the lowest reported by any fibre to date at their respective wavelength of operation, are combined with other unique properties offered by air-guiding fibres (and NANF in particular), the potential of this technology to transform numerous applications of high scientific and commercial interest becomes evident.

For example, wavelengths between 580 and 900 nm are often employed in the generation and use of quantum states and entangled photons that are essential for quantum computing, quantum networks, and quantum memories. The numerous approaches currently under study for use within such quantum systems exploit, among others, optical transitions in rare-earth-doped crystals, at 580 nm (Europium), 606 nm (Praseodymium), and 793 nm (Thulium)[56]; the transition wavelengths of alkali vapours at 780 nm or 795 nm (Rubidium) and 852 or 895 nm (Caesium)[57,58]; the NV (nitrogen-vacancy) and NE8 (nickel nitrogen complex) defect centres in a diamond around 637 nm and 800 nm, respectively[59]; and semiconductor (InAs) quantum dots embedded in GaAs, emitting around 900 nm, to name but a few.

Transmission of qubits through both short-distance applications, e.g., within quantum computers and memories, and medium to long-distance ones, e.g., for quantum teleportation and quantum key distribution experiments, relies on the availability of suitable optical fibres. The desire to exploit the existing all-solid fibre technology and its related optical infrastructure has prompted the development of frequency conversion methods to downshift photon energies and use the quantum systems at the minimum loss of conventional optical fibres, around 1550 nm[60]. However, clear advantages in terms of efficiency, complexity, and cost would arise if custom fibres operating with low loss directly at the source wavelengths could be employed. The use of hollow-core NANFs in these applications would not only bring a longer reach than is possible with all-solid fibre through a lower loss, but also a substantial reduction in other mechanisms that impair the transmission of optical qubits. It would enable, for example, orders of magnitude reduction in polarisation cross-talk and back-scattering over conventional fibres[35], as well as a reduced chromatic dispersion, nonlinearity, and sensitivity of the propagation time and phase to the external temperature[33,34]. Qubit transmission at wavelengths shorter than ~1000 nm would also allow the use of cheaper and higher performance silicon single-photon avalanche photodiodes (SI SPADs)[61], enabling the more widespread adoption of quantum key distribution devices and services[62].

At wavelengths around 850 nm the hollow core NANF could also be transformative for numerous other applications involving data communication, laser synchronisation, and time distribution. For example, it could offer an alternative to graded-index multi-mode fibres, for decades the most cost-effective solution for operation with VCSELs in low-cost, short-reach datacoms applications. The lower loss of NANF (1.45 dB km$^{-1}$ measured in the current work and <0.5 dB km$^{-1}$ potentially achievable with realistic structural improvements, as compared to 2.3 dB km$^{-1}$ for OM4/OM5 fibres[49]), combined with its lower dispersion (2–3 ps nm$^{-1}$ km$^{-1}$), large mode field diameter (20–25 μm), and effectively single-mode operation, could significantly increase the bandwidth-distance product of current short-distance data communications, unlocking further capacity improvement needed within hyper-scale datacentres.

The low dispersion and wide bandwidth (Fig. 4c), combined with a superior thermal stability[34], low nonlinear coefficient, and the recently demonstrated capacity to maintain pure polarisation states of NANFs[35], can also impact the delivery of ultra-short pulses from Ti:sapphire laser systems operating around 850 nm. This could allow, for example, passive synchronisation with unprecedentedly high stability and low timing jitter between independent lasers, for use in pump-probe spectroscopy[63], frequency metrology[64], and for precise timing distribution within large scale facilities, e.g., free electron lasers[65].

In the waveband between 1030 and 1090 nm where Nd:YAG lasers, Yb:YAG disk lasers, and Ytterbium fibre lasers are extensively used for cutting, welding, drilling, and marking, hollow-core fibres are already enabling power delivery at transmission distances and power levels not possible with solid fibres. For pulsed laser sources, the low nonlinearity and high damage threshold of HCFs have already enabled transmission of peak powers exceeding 10 s MW for ns/ps pulses[66] and GWs for fs pulses[37,67], which would cause catastrophic damage in an all-solid optical fibre due to self-focusing and dielectric breakdown of the glass[68]. Transmission of more than a kW of average power from a CW laser has also been reported in HCFs, although only over a few m[69]. The ultra-low loss reported here (0.52 dB km$^{-1}$ at 1064 nm, which could be reduced further following the ideas in Fig. 4d, e), could in principle enable kW power delivery in a single spatial mode over a significantly longer transmission range, potentially up to several km. This has the potential to extend the application areas for high power laser sources, enabling e.g., distribution of laser power to multiple workstations in a manufacturing plant[70], or deep subsurface rock drilling for exploration of oil, gas, and geothermal energy resources[71].

In conclusion, in this work, we have demonstrated an optical waveguiding technology that can outperform solid-core silica fibres at wavelengths of relevance to multiple key applications. We have reported three different NANFs designed to operate at 660, 850, and 1064 nm and achieving a loss of 2.85, 1.45, and 0.51 dB km$^{-1}$, respectively at those wavelengths. To the best of our knowledge, these losses represent not only a new record for hollow-core fibres, but crucially also an improvement (significant at 660 nm, more modest but measureable at 850 nm, within the measurement error at 1064 nm) over the best results ever

achieved with all-solid glass fibres. While these particular wavelengths were targeted here, shifting their operation to any other wavelength in the 600–1100 nm range would only require minimum tweaks in fabrication parameters, and NANFs with 0.28 dB km$^{-1}$ operating at 1550 nm have already been reported. Besides, the hollow core NANFs described in this work provide improved nonlinearity, dispersion, and environmental sensitivity as compared to conventional fibres. Clearly, NANF technology is still in its infancy, with considerable theoretical and technological progress still required to transfer its impact from the research lab to mainstream commercial applications. However, the results presented here, combined with the staggering recent progress in other areas[29,35] and with theoretical predictions indicating that they are still far from fundamental limits[25], show that hollow-core NANFs are ready to make dramatic contributions to several rapidly developing applications. For the first time in 50 years we might have finally identified an optical fibre technology with the potential to complement, and at least in the cases outlined above, replace all-solid silica fibres.

## Methods

**Fabrication**. Utilising experimental trials supported by simulations, we developed a fabrication technique and identified scalable fibre drawing parameters that allowed us to manufacture these fibres. The fibres were fabricated in a two-stage stack, fuse, and draw process, where cladding capillaries surrounding a central air core and consisting of 6 outer silica tubes with 6 nested inner tubes were stacked and fused inside a jacket tube and drawn to intermediate canes. The canes were then scaled down to fibre dimensions using a conventional cane-in-tube process. The core region and cladding tubes were pressurised throughout the fibre draw process, with the choice of pressure differentials guided by in-line fluid dynamics modelling[30]. The thicknesses of the 12 silica membranes surrounding the core were engineered to satisfy the anti-resonance conditions at the operational wavelength of interest[20,39]. In this work, we only preview and focus on the operational anti-resonance window of interest for each fibre, and have omitted other higher or lower transmission passbands[20,39].

**Optical attenuation measurements**. The optical attenuation of these fibres was measured using the cutback technique, described below. Keeping the NANF input end unaltered, three separate transmission traces from three different cleaves for the NANF output end were recorded for both long and short lengths. The average transmission of each length was then used to calculate the loss values and estimate the uncertainty level due to the different cleaves, shaded in grey in Fig. 2, and enumerated by the (±) values. Here we performed two types of measurements, single wavelength cutback only on *NANF-A*; and broadband measurements on all three NANFs. The setups used for the measurements vary based on the operational wavelength and are outlined as follows.

i. *NANF-A Single wavelength measurements:* With the purpose of minimising the cutback length, a free-space optics launch setup was prepared to excite predominantly the fundamental mode (FM) using a Bookham 1064 nm LC96A1060-20R laser diode module, emitting a single-mode output through a connectorised PM-980 fibre. A separate connectorised PM-980 fibre pigtail with a 90 degree angle cleave was used as launch fibre for the free-space launch setup, which consisted of two Thorlabs aspherical lenses: C240TME-C (focal length 8 mm) and C280TM-C (focal length 8 mm). The lenses were positioned to map the MFD of the PM-980 fibre to the MFD of the NANF FM (estimated to 22.4 μm). The power at the output of the NANF was measured using a Thorlabs S132C power metre.
Trials to identify a suitable cutback length for this setup were performed leading to a cut from 412 m to 1 m and gives an estimated FM propagation loss of 0.51 ± 0.07 dB km$^{-1}$.
Note that the positions of the input and output tip of the NANF were precisely monitored using cameras, to allow repeatable launch and detection conditions when testing multiple NANF samples with the setup. For example, repeated trials performed using a white light source, *cf.* broadband measurements below, of disconnecting and reconnecting the 412 m NANF with new input and output cleaves resulted in a power variation below ±0.02 dB over the low-loss range ~1020–1180 nm.

ii. *NANF-A Broadband measurements:* broadband measurements were performed with the same launch setup as the single wavelength measurement discussed above, with the PM-980 launch fibre connected directly to the white light source (WLS, Bentham WLS100). The NANF output spectrum was measured by butt-coupling the NANF output tip, deployed on a 1-m circumference bobbin at low tension, to a 25 μm core step-index multimode fibre (Thorlabs FG025LJA) connected to an optical spectrum analyser (ANDO 6315 A).

For the spectral loss measurements using the WLS, chromatic aberration in the lens-based coupling setup implies that the launch conditions will deviate from the optimum FM excitation as the wavelength diverges from the 1064 nm wavelength (for which the setup is optimised). Hence, a longer cutback length was required to compensate for this effect by providing sufficient attenuation of the increasing higher-order mode content away from 1064 nm. It was found that a length of 5 m was sufficient in the low-loss wavelength range of the NANF (<1 dB km$^{-1}$). However, the initial cutback test from 411 m to 5 m still resulted in an overestimate of up to ~0.2 dB km$^{-1}$ of the propagation loss compared to the single-wavelength measurement. This was attributed to the presence of residual light outside the single mode supported by the PM-980 launch fibre, caused by insufficient fibre length between the WLS and the launch tip. It was found that this residual light could be suppressed by tight coiling of the PM-980 fibre (about 30 turns of 0.5 inch diameter). The output spectrum was then measured again for the 5 m cutback sample and the 406 m length using identical launch and detection conditions. Taking the difference between the corresponding output spectra yields a good estimate of the propagation loss due to the repeatability of the launch conditions. The validity of the loss measurement is further supported by the excellent agreement with the single-wavelength measurement (within the error margins), as shown in Fig. 2.

iii. *NANF-B Broadband measurements:* for this fibre we used the Bentham WLS100 as the source, and an optical spectrum analyser (OSA, Yokogawa AQ-6315A, wavelength range 400–1750 nm) for data collection and analysis. The full length of the fibre of 1060 m was spooled on 1-m circumference bobbin at low tension, with 10 m of this length deployed loosely on the optical bench. A large-area single mode fibre (NKT LMA-25) was connected to the WLS using a bare fibre adaptor at one end, with the other butt-coupled to the to the input end of the NANF, to ensure a good single mode launch into the NANF. The output of the NANF was connected to the OSA using a bare fibre adaptor, where traces for three different end-face cleaves of both long and short fibre lengths were collected.

iv. *NANF-C Broadband measurements:* The full length of this fibre was deployed in two pieces, long piece of 569 m and short piece of 266 m, with each piece spooled on 1-m circumference bobbin at low tension. For the cutback measurements, the short piece of 266 m was connected to the Bentham WLS100 using a bare fibre adaptor and was butt-coupled to the longer fibre piece of 569 m using an optical fibre splicer. Three separate transmission spectra from three different end-face cleaves of the NANF were recorded using the OSA for the full length ~835 m, then, for the short length 266 m, while maintaining the launch input uninterrupted.

**Loss Simulations**. A commercial finite element method (Comsol Multiphysics®) was used to calculate the modes of the fibre. The geometry profile of *NANF-B* was reconstructed from Scanning Electron Microscope images (SEMs) of the fibre end-face. An in-house image processing code was developed to identify circles using an integrated circle finding routine followed by edge detection. The microstructure is built from circles identified in this way and their dimensions are conveyed to Comsol. The thickness of each of the capillaries is determined using glass volume conservation from the originally constructed stack of drawn tubes. It was found that a small modification to the thickness of the inner tubes was required to match the position of the experimental transmission window. This discrepancy was approximately 6% and is due to the change in the width of the bond between the microstructure capillaries and the jacketed glass during the draw which distorts that calculation.

A triangular mesh is used with 5 elements across the capillary width, and the largest element is no more than a fifth of the solved wavelength. The modes are solved at each wavelength in the range, stepping 5 nm.

The confinement loss of each mode is found from the imaginary part of the mode's effective index and surface scattering loss is determined from the normalised interface field intensity (F parameter). Both mechanisms are described in ref. [25].

Our calculations of microbending adopt the approximation whereby the NANFs are considered as effectively single moded. As a result, the theory developed by Peterman provides an appropriate approximation and gives the microbend loss as[41]:

$$\alpha_\mu = \frac{1}{2}(k_0 n_0 w_0)^2 \Phi\left(\frac{1}{k_0 n_0 w_0^2}\right) \qquad (1)$$

where $k_0 = 2\pi/\lambda$ is the free space wavenumber, $n_0 = 1$ the refractive index of the core, $w_0$ the mode spot radius, and $\Phi(.)$ the power spectrum of the random fibre axis curvature. For random stationary processes $\Phi$ typically decreases rapidly with spatial frequency. It follows therefore that at shorter wavelengths, microbending loss is higher because of the larger $k_0$, but also because at the smaller corresponding spatial frequency $(1/k_0 n_0 w_0^2)$ $\Phi$ becomes larger. By reducing the core size, microbending is reduced by virtue of a smaller $w_0^2$ which also results in a larger spatial frequency and thus a smaller $\Phi$. Since $\Phi$ results from random external loads applied to the fibre, effective methods for minimising their effects, first explored by Gloge[42], include the use of stiffer fibres (i.e., larger glass diameter) and softer coating materials, like in any other type of optical fibre. For our calculations, we

used a curvature spectrum Φ equivalent to isolated perturbations as described in ref.[72] and introduced a free parameter accounting for the strength and number of such perturbations per unit length to fit the computed loss to experiments.

The radial Poynting vector shown in red around the perimeter of the structure in Fig. 4a is the azimuthally resolved radial component of the time-averaged Poynting vector calculated on the inside edge of the PML. The radial Poynting vector at the edge of the solved domain can be integrated along the outer boundary of the fibre and normalised by the total power propagating in the axial Poynting vector to provide the confinement loss. In this way, local leakage loss can be quantified through a line integral along sectors on this outer boundary, as given in the labels of Fig. 4a. The figure shows that approximately 75% of the power leaks through 50% of the cladding but no single leakage feature is particularly significant.

## Data availability

All data supporting this study are available from the University of Southampton repository at https://doi.org/10.5258/SOTON/D1403. Other findings of this study are available from the corresponding author on reasonable request.

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

## Acknowledgements

This work has received funding from the European Research Council (ERC) through the project Lightpipe (grant agreement no. 682724). G.T.J. and E.N.F. acknowledge support from the Royal Academy of Engineering through personal research fellowships. The authors gratefully acknowledge insightful conversations with Professor Sir David Payne and Professor David Richardson at the University of Southampton. We also acknowledge Dr Seyed Reza Sandoghchi for collecting the mode field image in Fig. 1.

## Author contributions

G.T.J., E.N.F., and F.P. designed the fibres. Y.C. and J.R.H. developed initial stage pre-form fabrication techniques prior to fibre fabrication. H.S., Y.C., T.D.B., and I.A.D. fabricated the fibres. H.S. and H.C.M. measured the optical loss. H.S. and I.A.D. performed SEM. F.P., Y.C., H.S., and G.T.J. wrote the manuscript. F.P. provided overall technical leadership across all aspects of the research.

## Competing interests

The authors declare no competing interests.
