## [Peer Review File · Nature Communications]

REVIEWER COMMENTS

Reviewer #1 (Remarks to the Author):

Demonstration of loss in hollow-core fiber below Rayleigh scattering limit in solid-core fibers was first achieved using a co-joined tube negative curvature fiber with 4.9dB/m at $\sim 560\text{nm}$ (H6 in Fig. 3, ref.32). This current work demonstrated loss below Rayleigh scattering limit in solid-core fibers at wavelength below 700nm. The results are comparable to that achieved in [32]. The authors used nested tube design, a somewhat different but fundamentally similar design. At longer wavelength, the loss is similar to record loss achieved in solid-core fibers. The authors does show much improved control and consistency of their fibers; the dominating loss is from microbending loss; and much lower loss is expected if microbending loss can be lowered. Hollow-core fiber does have narrow bandwidth where the loss is low, especially comparing to solid-core fibers, but has much lower dispersion in the short wavelengths. Their applications in short wavelength regime are not entirely clear at this point. Liang Dong

Reviewer #2 (Remarks to the Author):

The manuscript "Hollow core optical fibres with comparable attenuation to silica fibres between 600 and 1100 nm" presents hollow core fibres with losses comparable or lower than Rayleigh scattering loss in solid glass fibers. This manuscript shows excellent results in both experiments and simulations. The start of pull (SOP) and end of pull (EOP) cross-sections of the fibre show uniformity along the as-drawn fibre length.

The Fig. 4a shows loss according to the azimuthally resolved radial component of the time averaged Poynting vector calculated on the inside edge of the PML. The calculation captures the leakage of the fiber from the antiresonant structures. However, the loss values provided are all substantially lower than the confinement loss indicated by the blue dashed curve ($\sim 0.2\text{ dB/km}$) in Fig. 4b. The reviewer would think that the average value or the integrated value of the loss around all angles in Fig. 4a should be comparable to the blue dashed curve ($\sim 0.2\text{ dB/km}$) in Fig. 4b. Please provide a few sentences to explain this minor point.

I recommend publication of this paper in Nature Communications.

Reviewer #3 (Remarks to the Author):

The authors have submitted a paper that demonstrates continued improvement of their fiber designs. The established NANF design has proven to have the lowest loss of any hollow core fiber design and a very wide window and in this paper the authors demonstrate along with the low loss and relatively wide windows of operation, the fibers now have lower loss than conventional all solid silica fibers at wavelengths shorter than the lowest loss telecommunications window.

The authors present three fibers that have consistently good optical performance and demonstrate their skill and knowledge in choosing a higher order window for the lowest wavelength fiber. This is to obtain good structural uniformity with the thicker struts compared to the required thinner struts in the second order window.

The authors are very open about the measurement methodology and the structure of the fiber which the readers will appreciate very much. The outer diameter of the fiber along with coating diameters (and coating material) should also be included as these are key parameters in the microbending issue (which the authors have disclosed). In lines 193-196 the authors discuss a method to mitigate the microbend losses at shorter wavelengths. I am not aware of any paper that describes the issues and performance of the fiber with regards to microbend. To further improve the paper a short discussion on microbending would be beneficial. This is key to how the fiber could be used, and in the latter part of the paper the authors describe many applications for use.

This paper describes work done on the fiber design and fabrication to reduce the loss in HCF below the Rayleigh limit of silica. It's important to note that the authors are not the first to publish this (they also say this and fully reference the work) but in the context of this unique design presented in this paper, which has significant opportunities for even further growth, this is an important result. I'm not sure the conjoined tube idea can be taken further.

REVIEWER COMMENTS

Reviewer #1 (Remarks to the Author):

Demonstration of loss in hollow-core fiber below Rayleigh scattering limit in solid-core fibers was first achieved using a co-joined tube negative curvature fiber with 4.9dB/m at ~560nm (H6 in Fig. 3, ref.32). This current work demonstrated loss below Rayleigh scattering limit in solid-core fibers at wavelength below 700nm. The results are comparable to that achieved in [32]. The authors used nested tube design, a somewhat different but fundamentally similar design. At longer wavelength, the loss is similar to record loss achieved in solid-core fibers. The authors does show much improved control and consistency of their fibers; the dominating loss is from microbending loss; and much lower loss is expected if microbending loss can be lowered. Hollow-core fiber does have narrow bandwidth where the loss is low, especially comparing to solid-core fibers, but has much lower dispersion in the short wavelengths. Their applications in short wavelength regime are not entirely clear at this point. Liang Dong.

We thank Prof Dong for his comments. We take his final comment about short wavelength applications as an observation rather than a request to modify the manuscript. We indeed agree with him that out the 3 wavebands identified in the paper those around 1060 and 850nm seem to be the closest to achieving impact in practical applications. The most appealing application for shorter wavelengths is arguably for quantum communications, as we extensively discuss in the paper, but certainly considerable more work will be needed before impact in this direction is achieved.

Reviewer #2 (Remarks to the Author):

The manuscript "Hollow core optical fibres with comparable attenuation to silica fibres between 600 and 1100 nm" presents hollow core fibres with losses comparable or lower than Rayleigh scattering loss in solid glass fibers. This manuscript shows excellent results in both experiments and simulations. The start of pull (SOP) and end of pull (EOP) cross-sections of the fibre show uniformity along the as-drawn fibre length.

We are grateful for the reviewer's praise of our results. Indeed, we are also pleased with the results achieved and the uniformity of our fibres.

The Fig. 4a shows loss according to the azimuthally resolved radial component of the time averaged Poynting vector calculated on the inside edge of the PML. The calculation captures the leakage of the fiber from the antiresonant structures. However, the loss values provided are all substantially lower than the confinement loss indicated by the blue dashed curve (~0.2 dB/km) in Fig. 4b. The reviewer would think that the average value or the integrated value of the loss around all angles in Fig. 4a should be comparable to the blue dashed curve (~0.2 dB/km) in Fig. 4b. Please provide a few sentences to explain this minor point.

We thank the reviewer for his/her valuable comments. The integrated value should sum to the same as the dashed curve. The total radial loss (by integration around the whole fibre) is equal to the confinement loss at 850nm, but the graphic in Fig. 4a only shows the more significant leakage zones to highlight how some structural features are more significant than others. Summing those labels gives a total of: $0.055 + 0.047 + 0.027 + 0.021 + 0.019 + 0.019 = 0.188\text{dB/km}$. Indeed, about 75% of the leakage occurs around 50% of the perimeter made up of a number of small leakage features. To make the manuscript clearer, we have made the following changes:

The body text referring to Fig 4a on page 11 was:

Fig. 4(a) plots the azimuthally-resolved rate of power leakage through a cross-section of the fabricated fibre at 850 nm, and shows that due to a good symmetry and small inter-tube gaps the rate of leakage is low for every azimuthal position, contributing to only 0.17 dB/km.

We changed to:

Fig. 4(a) plots the azimuthally-resolved rate of power leakage through a cross-section of the fabricated fibre at 850 nm, and shows that due to a good symmetry and small inter-tube gaps the rate of leakage is low for every azimuthal position, with no geometric features contributing more than 25% to leakage loss.

The caption of Fig.4a now reads:

(a) Fundamental mode at 850 nm with contours of longitudinal Poynting vector, and azimuthally resolved power leakage (radial Poynting vector) is visualised as a red external curve - local regions of high leakage are highlighted and quantified in the labels.

We also modified the words in the simulations section of the methods as follows:

The radial Poynting vector shown in red around the perimeter of the structure in Fig. 4(a) is the azimuthally resolved radial component of the time averaged Poynting vector calculated on the inside edge of the PML. The radial Poynting vector at the edge of the solved domain can be integrated along the outer boundary of the fibre and normalised by the total power propagating in the axial Poynting vector to provide the confinement loss. In this way, local leakage loss can be quantified through a line integral along sectors on this outer boundary, as given in the labels of Fig. 4(a). The figure shows that approximately 75% of the power leaks through 50% of the cladding but no single leakage feature is particularly significant.

I recommend publication of this paper in Nature Communications.

We highly appreciate the reviewer's recommendation.

Reviewer #3 (Remarks to the Author):

The authors have submitted a paper that demonstrates continued improvement of their fiber designs. The established NANF design has proven to have the lowest loss of any hollow core fiber design and a very wide window and in this paper the authors demonstrate along with the low loss and relatively wide windows of operation, the fibers now have lower loss than conventional all solid silica fibers at wavelengths shorter than the lowest loss telecommunications window.

The authors present three fibers that have consistently good optical performance and demonstrate their skill and knowledge in choosing a higher order window for the lowest wavelength fiber. This is to obtain good structural uniformity with the thicker struts compared to the required thinner struts in the second order window.

We highly appreciate the reviewer's positive comments.

The authors are very open about the measurement methodology and the structure of the fiber which the readers will appreciate very much. The outer diameter of the fiber along with coating diameters (and coating material) should also be included as these are key parameters in the microbending issue (which the authors have disclosed). In lines 193-196 the authors discuss a method to mitigate the microbend losses at shorter wavelengths. I am not aware of any paper that describes the issues and performance of the fiber with regards to microbend. To further improve the paper a short discussion on microbending would be beneficial. This is key to how the fiber could be used, and in the latter part of the paper the authors describe many applications for use.

We thank the reviewer for his/her comments on the microbending and for the suggestion to add a short discussion about it.

In response to this request, we have added the following discussion in the Methods section:

Our calculations of microbending adopts the approximation whereby the NANFs are considered as effectively single moded. As a result, the theory developed by Peterman provides an appropriate approximation and gives the microbend loss as⁴¹:

$$\alpha_{\mu} = \frac{1}{2}(k_0 n_0 w_0)^2 \Phi \left(\frac{1}{k_0 n_0 w_0^2} \right) \quad (1)$$

where $k_0 = 2\pi/\lambda$ is the free space wavenumber, $n_0 = 1$ the refractive index of the core, w_0 the mode spot radius and $\Phi(\cdot)$ the power spectrum of the random fibre axis curvature. For random stationary processes Φ typically decreases rapidly with spatial frequency. It follows therefore that at shorter wavelengths, microbending loss is higher because of the larger k_0 , but also because at the smaller corresponding spatial frequency ($1/k_0 n_0 w_0^2$) Φ becomes larger. By reducing the core size, microbending is reduced by virtue of a smaller w_0^2 which also results in a larger spatial frequency and thus a smaller Φ . Since Φ results from random external loads applied to the fibre, effective methods for minimizing their effects, first explored by Gloge⁴², include the use of stiffer fibers (i.e., larger glass diameter) and softer coating materials, like in any other type of optical fibre. For our calculations, we used a curvature spectrum Φ equivalent to isolated perturbations as described in ref.⁷² and introduced a free parameter accounting for the strength and number of such perturbations per unit length to fit the computed loss to experiments.

Additionally, in the main text we have cited the Gloge and Peterman works in the above-mentioned lines, and inserted the following statement regarding the coatings used:

Additionally, the fibres were coated with commercially available low modulus UV- curable polymers.

This paper describes work done on the fiber design and fabrication to reduce the loss in HCF below the Rayleigh limit of silica. It's important to note that the authors are not the first to publish this (they also say this and fully reference the work) but in the context of this unique design presented in this paper, which has significant opportunities for even further growth, this is an important result. I'm not sure the conjoined tube idea can be taken further.

We highly appreciate the reviewer's very positive comments on the future of our NANF design.

REVIEWERS' COMMENTS

Reviewer #2 (Remarks to the Author):

publish

Reviewer #3 (Remarks to the Author):

Thank you for the nice replies and amendments to the manuscripts to help with understanding how you see the principles of operation. I appreciate the time put in to this.